# Usage of Mobile Applications or Mobile Health Technology to Improve Diet Quality in Adults

**DOI:** 10.3390/nu14122437

**Published:** 2022-06-12

**Authors:** Alan Scarry, Jennifer Rice, Eibhlís M. O’Connor, Audrey C. Tierney

**Affiliations:** 1Department of Biological Sciences, University of Limerick, V94 T9PX Limerick, Ireland; eibhlis.oconnor@ul.ie; 2Department of Physical Education and Sports Science, University of Limerick, V94 T9PX Limerick, Ireland; jennifer.rice@ul.ie; 3Health Research Institute, University of Limerick, V94 T9PX Limerick, Ireland; 4APC Microbiome Ireland, University College Cork, T12 K8AF Cork, Ireland; 5School of Allied Health, University of Limerick, V94 T9PX Limerick, Ireland; audrey.tierney@ul.ie; 6Health Implementation Science and Technology Cluster, Health Research Institute, University of Limerick, V94 T9PX Limerick, Ireland; 7Department of Dietetics, Human Nutrition and Sport, La Trobe University, Melbourne, VIC 3086, Australia

**Keywords:** Diet*, Nutri* technology, app, mobile app, M-health, E-health, health technology, mobile health, technology and health software

## Abstract

The use of mobile applications for dietary purposes has dramatically increased along with the consistent development of mobile technology. Assessing diet quality as a dietary pattern or an indicator across key food groups in comparison to those recommended by dietary guidelines is useful for identifying optimal nutrient intake. This systematic review aims to explore mobile applications and their impact on the diet quality of the user. The electronic databases of The Cumulative Index to Nursing and Allied Health Literature (Cinahl), The American Psychological Association’s (APA Psycinfo), and PubMed were systematically searched for randomised and non-randomised controlled trials to retrieve papers from inception to November 2021. Ten studies with 1638 participants were included. A total of 5342 studies were retrieved from the database searches, with 10 articles eligible for final inclusion in the review. The sample sizes ranged from 27 to 732 participants across the included studies, with 1638 total participants. The ratio of female to male participants in the studies was 4:1. The majority of the mobile applications or M-health interventions were used to highlight dietary health changes (six studies), with the remainder used to reduce weight or blood sugar levels (four studies). Each study used a different measure to quantify diet quality. Studies were either assessed by diet quality scoring or individual dietary assessment, of the ten studies, six studies reported an improvement in diet quality following diet-related mobile application use. Mobile applications may be an effective way to improve diet quality in adults; however, there is a need for more targeted and longer-term studies that are expressly designed to investigate the impact using mobile applications has on diet quality.

## 1. Introduction

Modifiable risk factors including diet are associated with the development and progression of cardiovascular disease (CVD), type two diabetes mellitus (T2DM) and non-alcoholic fatty liver disease (NAFLD), along with links to cancers, obesity and other all-cause mortalities [1]. Much of this research reports that a commitment to a high-quality diet improves health outcomes [2]. Over time, the assessment of diet quality has seen the emergence and implementation of several scoring systems or measurement indices. Diet quality can be calculated by scoring how closely food patterns align with national dietary guidelines and on how diverse and healthy choices are made [3]. According to the Healthy Eating Index (HEI), diet quality is scored on components including: total fruit (5 points), whole fruit (5 points), total vegetables (5 points), dark green and orange vegetables and legumes (5 points), total grains (5 points), whole grains (5 points), milk (10 points), meat and beans (10 points), and oils (10 points). A higher diet quality score aligns with better health [4,5]. The benefit of more advanced scoring methods of diet quality is that they allow both protective and unfavourable dietary patterns to be identified [3]. Mobile (M)-Health technology is described by the World Health Organisation as medical and public health practice supported by the use of mobile devices, patient monitoring devices, personal digital assistants (PDAs), and other wireless instruments [6]. Mobile devices such as phones, smartphones, and tablets have multiple features that facilitate dietary management and are some of the more prevalent technologies employed [7]. Globally mobile health interventions utilising smartphone applications are advancing rapidly to facilitate and provide practical health-promoting approaches [8]. The use of M-health in healthcare settings extends health surveillance and includes the categorising and determining of ‘risk factors’ and ‘at-risk groups’ that are then considered suitable for intervention. Health-related data may be readily and repeatedly gathered from users’ mobile devices whenever they access a relevant mobile application [9]. Mobile phones can offer various options in a health setting, from text messages, which can be used for one-way information delivery or counselling support, to the more advanced smartphones with several functions. Indeed smartphones offer an increased capacity (beyond standard internet connectivity) for dietary monitoring allowing the user access to features for self-monitoring of dietary behaviors [10]. This technology has proven useful in the implementation of lifestyle changes, disease prevention, managing conditions such as diabetes and promoting behavioural adaptations [3]. Diet and nutrition M-health applications (apps) are among the fastest-growing area of health promotion apps [10]. 

Diet-tracking apps have considerable popularity, with some downloaded up to 50 million times (based on MyFitnessPal for google Play Store, April 2017) [11]. Dietary mobile applications have a role in helping individuals lose weight, managing chronic conditions and understanding dietary patterns and the quality of nutrient intakes [11]. Typically M-health or Electronic (E)-health mobile applications support users in monitoring their food and exercise and provide users with Appendix A about these actions such as calories consumed or utilised and predictions on weight loss goals [5]. This information is displayed digitally and is further supported by ‘nudges’ which can prompt the user to perform an activity such as drink a glass of water, exercise or eat a healthy snack [5].

Mobile apps offer both the user and the clinician the ability to collect health information and deliver real-time feedback outside the clinical setting. Research into the use of M-health technology highlights that clinicians prefer integrated technology, which allows them to concentrate on patient-specific needs [12]. However, despite this, most of the research conducted in this area relates to app rating ability or patient success, such as weight management or managing dietary conditions such as type two diabetes or other all-cause morbidities measured by weight reduction, decreased body mass index (BMI) or HbA1c [7]. With respect to diet and dietary behaviours, numerous mobile apps are available, however, weight and eating behaviour management are the most sought-after changes through self-monitoring [9].

According to Stehr et al. (2020), research on nutrition apps thus far has concentrated on evaluating their effectiveness, which has led to variable results [13]. While a change towards increased diet quality or attainment of diet-related goals is positive, there is concern that mobile applications and their use in dietary health can promote negative dietary habits; an overdependence on the application may affect the long-term benefits to the user [14]. Studies have shown that weight loss achieved with the use of technology may promote unhealthy eating behaviours such as orthorexia and anorexia [15,16]. With applications so freely available, the consumer can access dietary advice without having any contact with a dietitian or registered nutritionist. This means the user is solely reliant on the mobile app for dietary advice, which may vary in terms of scientific rigor. In selecting which dietary applications to use, the consumer may be relying solely on subjective consumer ratings and reviews whereby the consumer becomes restricted in their choice of application as the content is produced with a large audience intention and not tailored to meet specific individual behavioural needs [15,16].

Whilst weight loss is the most desired outcome for many app users, weight loss is not easy to maintain or sustain long term. Looking at improving diet quality in the absence of weight loss is a more sustained approach to improving health and diet-related conditions [17]. Although multiple systematic reviews have explored dietary applications and how they enable the user to monitor their diet with or without a clinician and focus on the promotion of weight loss, the focus of this review is on the impact of mobile applications on diet and nutritional quality

## 2. Materials and Methods

### 2.1. Search Strategy

This systematic literature review’s study design and analysis are in accordance with the preferred reporting items for systematic reviews and meta-analysis guidelines (PRISMA) (Appendix A) [18]. Electronic database searches were carried out in the Cumulative Index to Nursing and Allied Health Literature (APA, Psycinfo), and PubMed in October 2020. Key terms for diet were combined with terms for mobile health technology or health software: (“Diet*, Nutri* Technology, App, Mobile App, M-Health, E-Health, Health Technology, Mobile health Technology, and Health software”). See Appendix A for full search strategy. The literature search was limited to randomised controlled trials (RCTs), non-randomised controlled trials, interrupted time series, controlled before-and-after studies, and cohort studies. Unpublished studies (e.g., conference abstracts, trial protocols, surveys) were excluded.

Title, abstract, and keywords were searched in databases and Google scholar; in addition, reference lists and citations of relevant located articles were searched. Articles reviewed were limited to English-language published in peer-reviewed scientific journals between 2010 and October 2020. Furthermore, due to technology development for dietary intake assessment since 1st January 2010, searches were limited to articles published after this time [19]. This review is registered with PROSPERO international prospective register of systematic reviews (CRD42021231709). Meta-analysis of outcomes was not possible due to the heterogeneity of variables assessed, and therefore a descriptive narrative review was conducted. The protocol study for this review is published and available at https://hrbopenresearch.org/articles/5-1 (accessed on 5 May 2022) [20].

### 2.2. Study Selection

Studies were selected through a two-stage process. Firstly, two reviewers (AS, JR) independently examined titles and abstracts using Rayyan, a web application for systematic reviews [21]. When a selection could not be completed based on this information alone, the article was included for the next stage. The first reviewer (AS) reviewed all articles, and 20% of the papers were assessed by reviewer two (JR). If divergences arose, there was additional discussion with other reviewers (AT, EOC). However, this process was not necessary during the study selection process.

### 2.3. Eligibility Criteria

Studies included in this review investigated adults over the age of 18 years and include the use of mobile health technology apps or other M-health interventions, including technology used for M-health such as mobile devices that have cellular communication capabilities allowing wireless interaction. Studies focused on diet quality as an outcome measure were to be included. Comparator groups were considered if there were control or alternative interventions to the use of mobile health technology if they also assessed diet quality. 

Primary outcomes of interest included change in diet quality. Due to the heterogeneity of the studies included, the parameters assessed for diet quality differed between studies. Using the metrics provided by each study and the information derived from nutrients, food or food groups, or diet quality scoring, an assessment was made if technology impacted diet quality within the study.

Secondary outcome measures such as weight change were also examined if included, but the reporting of these measures was not a requirement for inclusion in this review. All studies that met the eligibility criteria were assessed by reading full text. Discrepancies in eligibility of studies between the reviewers (AS, JR) were discussed with additional experienced reviewers (AT, EOC) until consensus was reached. Papers excluded from this review were non-human studies, studies relating to children, articles not available in the English language, articles not reporting primary outcomes of interest. 

### 2.4. Data Extraction and Outcomes

Data extraction was carried out on all articles that met the inclusion criteria. Information regarding the year, geographical location, study design, gender and age, duration of intervention, participant BMI, weight status (at the start and end of the study if available), mobile application used in the study, the intervention and, control details (if available), measures of diet quality and other outcomes were extracted. The use of independent third parties was not necessary, as the two primary assessors (AS, JR) reached a consensus regarding the data extraction of all articles.

### 2.5. Quality Assessment

All studies included were critically appraised by two independent authors (AS and JR) using the Cochrane risk-of-bias assessment tool and the Risk Of Bias In Non-randomised Studies—of Interventions (ROBINS) [22,23]. This tool assesses random sequence generation, allocation concealment, the blinding of participants and personnel, blinding of outcome assessment, incomplete outcome data addressed, selective reporting, and other biases [22].

According to the criteria, evaluations are scaled and scored as a low, unclear, and high risk of bias. A study with a low risk of bias needs to fulfil more than six of the listed items and have no critical difference between the start and end of the study [22,24] defined as a dropout greater than 50% or a statistically significant difference between groups at baseline indicating unsuccessful randomisation [22]. Any disagreements relating to the interpretation of the criteria were resolved through discussion. Third-party reviewers (AT, EOC) were available to arbitrate but were not required. 

## 3. Results

### 3.1. Study Selection

The initial electronic and manual search resulted in the retrieval of 5342 records. After the removal of 2347 duplicates, 2995 papers remained. Screening against eligibility criteria, 97 records were considered potentially eligible for the title and abstract screen. Ten studies fulfilled the inclusion criteria after reviewing full-text articles and independent assessment by a second reviewer (JR). The PRISMA flowchart is detailed in Figure 1 [18].

Of the 97 articles removed at the full-text screening stage, 79 studies did not meet the inclusion criteria. Forty studies did not clearly show a link between the use of mobile applications and diet quality. The primary focus of the ten articles was on the use of mobile applications or the comparison to other mobile applications rather than their use. Ten additional papers focused on the use of mobile applications for weight loss. Eight papers did not track diet consistently during their study or measure the effect of mobile technology. Six studies were protocol papers, and at the time of screening, no results were available. A further six studies were conference papers, which were excluded. Three papers were beyond the limitations of the study (papers no 48, 56 and 77 all included children under the age of 18, a population not of interest for this review.) Reasons for exclusion of all studies are documented in the PRISMA flowchart detailed in Figure 1 [24].

### 3.2. Study Characteristics

Ten studies met the inclusion and eligibility criteria for this review. There were seven randomised controlled trials (RCT) [25,26,27,28,29,30,31], the remaining three studies were a short report [32], a pilot study [32], and a feasibility study [33]. All studies were published in English. Five of the ten studies used definitions of diet quality, which corresponded closely to standard and widely accepted definitions [25,26,27,28,29,32]. Of the remaining five studies, two used metrics for measuring diet quality (a quality scoring of food items by food group according to the Australian Guide to Healthy Eating standard servings (AGHE) Kerr et al. 2016 and a dietary risk score (DRS), Van Djik et al. 2019). The final three studies focused on weight loss and dietary improvement as the primary outcome measure but used mobile applications and reported an improvement in the quality of the participant’s diet in line with accepted definitions of diet quality [27,33,34]. This systematic literature review takes the term diet quality from the Healthy Eating Index approved method, scoring diet quality based on its components. Additional research is incorporated regarding diet quality scores and weight [4,5]. The majority of studies were from the USA [25,26,27], Australia [26,28,32] or Europe [31,33], with a single study in India [30], Korea [34], and Israel [29]. 

### 3.3. Participants and Interventions

The total number of participants studied was 1638 across the 10 included studies. The sample sizes under investigation varied, from 732 [30] participants in an RCT to 27 participants in a feasibility study [33]. Nine studies had majority female participation, with three of those studies including female participants exclusively. While only one paper had a majority of male participation, no study included male participants exclusively. In all 10 papers, participants used diet-related mobile applications or M-health interventions. The applications reviewed in this study include My Fitness Pal, Easy Diet app, the mobile food record (mFR), The Lose It mobile application, Lifestyle Advice Plus, e-balance, AiperMotion 500, The Health-On mobile application, The Smart eating app and The Smarter Pregnancy program. (See Table 1).

### 3.4. Risk of Bias within Studies

The quality assessment of studies using the Cochrane risk of bias tool is illustrated in Figure 2. Based on our assessments, only one study was found to be of high risk of bias for most domains. The primary source of bias identified across studies was the blinding of the study outcome from participants. Three studies were identified as being at a high risk of bias as they did not meet the assessment criteria. Based on these assessments, only seven studies were found to be of low risk of bias for most domains. We included all studies in this review, including those deemed to be of high risk of bias as reporting was descriptive in nature and excluding them could alter the results and conclusions of this review.

### 3.5. Study Outcomes

A meta-analysis of outcomes was not possible due to the heterogeneity of variables assessed. Each study used a different mobile application and each study used a different measure to quantify diet quality, therefore data were described narratively. Diet quality was the focus of this research, and this was highlighted in different ways in the 10 studies examined in this review.

Six of the studies included directly linked mobile applications, participant recording of diet, and improvement of diet quality. In one of the studies there was no improvement in diet quality, however, an improvement in urinary sodium levels, which was the primary aim of this study, was reported [25]. Both Ambrosini et al. (2018) and Dodd et al., (2017) did not report a link between the use of mobile applications and improved diet quality [28,32].

Ambrosini et al. (2018) tested the feasibility of using smartphone technology for dietary assessment over using 24 h recalls, testing mean (SD) daily intakes of total energy (MJ), the proportion of energy (%E) from macronutrients, elected nutrient densities selected in this study as indicators of diet quality (fibre g/MJ) or marginal population intakes (calcium mg/MJ; iron mg/MJ) from each technique were estimated for evaluation [32]. During the study participants were required to complete a four-day food diary using mobile applications and separately complete two 24 h recalls which were given to participants over the phone before finally completing a questionnaire about both processes. Both the four-day food diary and 24 h recalls took place over a seven-day period. In the study by Ambrosini et al. (2018), the quality of the participant’s diet was measured by average energy ratios from macronutrient fibre, iron, and calcium densities from the app and the 24 h recalls [32]. The study, using mean daily nutrient intakes from both the mobile application and dietary recall, indicated no distinct differences between the app and the 24 h recalls for protein (protein intake (% E) mean ± SD average of RFD app 19.7 ± 4.9), fat (Total fat (% E) intake mean ± SD average of RFD app 34.0 ± 6.2), saturated fat (saturated fat (% E) mean ± SD average of RFD app 12.0 ± 2.8) carbohydrate (carbohydrate intake (% E) intake mean ± SD average of RFD app 39.8 ± 8.0), and iron density (iron density (mg/MJ energy intake) intake mean ± SD average of RFD app 1.6 ± 0.5); however, added sugar intake (added sugar intake (% E) mean ± SD average of RFD app 5.0 ± 3.4) was higher in those in the 24 h recalls, which may be due to underreporting, which was higher in the 24 h recalls as noted by the study [32]. 

Ipjian et al. (2017) examined, in a pilot study, the impact smartphone technology could have in facilitating dietary change in healthy adults, with the main focus being on diet quality and its effect on sodium intake. Diet quality was assessed via the Rapid Eating and Activity Assessment for Patients (max score 39 with a higher number indicating better diet quality). Diet quality scores were similar between the intervention and control groups at baseline [25]. Participants were split into two groups using either a mobile application or a journal to record and measure the quality of their diet. Although diet quality scores were comparable across both the intervention and control groups throughout the study, baseline values for urinary sodium excretion decreased in participants in the app group only when compared to the journal group (−838 ± 1093 versus +236 ± 1333 mg/24 h-*p* = 0.010) [25]. Study adherence was calculated as the number of days each participant recorded a whole food record(%) [25]. Although the 4-week study adherence did not vary between groups, there was a trend towards increased adherence in the app group compared with the journal group (92.1 ± 16.2% versus 82.1 ± 25.1%, respectively; *p* = 0.098) [25]. A secondary finding was the importance of app usability, stating that the participants who used a smartphone app reported a greater degree of tracking satisfaction than their equivalents who used the paper-and-pencil method [25]. 

Kerr et al. (2016) assessed diet quality using a Mobile Food Record (MFR app). This mobile method allowed participants to take image recordings of all items of food and drink consumed. The intervention was a randomised control trial with three arms, all of which used the mobile application: (A) dietary feedback and weekly text messages, group (B) received dietary feedback only and group (C) was the control group. Dietary intake was assessed using the mFR where participants captured images of foods and beverages consumed over 4 days at baseline and post-intervention [26]. An experienced reviewer evaluated the 4-day mFRs employing a scoring system to evaluate the quality of food commodities by food group (servings of fruits, vegetables and energy-dense nutrient-poor (EDNP food) and beverages according to the Australian Guide to Healthy Eating standard servings (AGHE)) [26]. Diet quality was measured in reference to portion size: for each participant, an average daily serving was estimated for fruits, vegetables, sugar-sweetened beverages, energy-dense nutrient-rich foods and alcohol [26]. Both intervention groups used the mFR but intervention group two did not receive additional text message support. The effect of the intervention was a reduction in energy-dense, nutrient-poor foods (intervention group (A) −0.8 ± 0.2 *p* < 0.001 intervention group (B), −0.8 ± 0.2 *p* < 0.001 and control −0.5 ± 0.2) and an increase in fruit servings in all participants (intervention group (A), −0.2 ± 0.1 *p* = 0.03) and vegetables servings in female participants (intervention group (A), 0.4 ± 0.2 *p* = 0.01, intervention group (B), 0.5 ± 0.1 *p* < 0.001, control group, 0.4 ± 0.2 *p* = 0.03) [26]. The results highlight improved diet quality with reduced energy-dense, nutrient-poor foods and increased portions of fruits and vegetables for daily consumption. 

Wharton et al. (2014) examined the impact the “Lose It” mobile application had on participants’ diet quality. There were three intervention groups; two used mobile applications to record dietary intake and the final intervention group used a traditional paper-and-pencil method. There was no difference in the baseline Healthy Eating Index Scores (HEI) between the groups; however, the resulting HEI scores declined somewhat (−6%) in the app group and increased somewhat in the other groups (+3% and +9% for the memo group (ME) and paper group (PA) groups, respectively) (*p* = 0.29; effect size = 0.089) [27]. While it was noted that the App users more consistently entered whole days of dietary data compared with the paper-and-pencil group, (43.0 ± 2.5 and 30.7 ± 4.6 days, respectively; *p* = 0.024; effect size = 0.153) [27], this study showed no significant difference when using mobile technology for improving diet quality. 

The study by Dodd et al. (2017) evaluated smartphone applications to provide lifestyle advice to pregnant women. The intervention study examined both the use of a mobile application and lifestyle advice, while the control group only received lifestyle advice. This study also used the HEI as a measure of diet quality [28]. At baseline assessment, the HEI score was comparable between the two treatment groups [28]. However, post-intervention it was noted that the addition of the smartphone application was not linked with any statistically notable variations in HEI score [28]. All participants in the trial showed increases in consumption of milk and whole grains and a decrease in sodium consumption over the trial [28]. There were no statistically significant differences between the two treatment groups after the trial; however, it must be noted that the use of mobile applications was low (31%) which may have influenced the findings [28]. 

Naimark et al. (2015) evaluated the significance of a newly designed web-based application (E-Balance) to encourage a healthy lifestyle and to educate adults on such lifestyles with face-to-face sessions, emails and web-based presentations delivered to the intervention group. The control group was only informed to “continue living a healthy lifestyle” [29]. The mobile application included components to improve usability, feedback tailored to the individual user according to the Dietary Reference Intake (DRI) recommendations, a large Ministry of Health food database, and a physical activity database [29]. The study measured diet quality based on guidelines from a 16-item questionnaire based on Parmenter’s general nutrition knowledge questionnaire for adults using an online self-reported questionnaire [35]. The study noted that the app users increased their diet quality score significantly at the end of the study from 67 (SD 9.8) to 71 (SD 7.6; *p* < 0.001) [29]. The success score was greater among the app group (68%) compared with 36% in the control group (*p* < 0.001) Success was defined as decreasing or preserving the original weight, period of physical activity equivalent to or more than 150 min per week, a score of more than 70 points on the nutrition and quality of diet questionnaires [29]. The improvement in diet quality scores was no different between light and heavy users of the mobile app. This study highlighted the importance of application usability, and noted high usability, increased nutritional knowledge, duration of physical activity, and reduced body weight [29]. 

Kaur et al. (2020) examined the effectiveness of an information technology-enabled ‘SMART eating’ health promotion intervention [30]. This intervention was aimed at behaviour change using a multi-channel interaction approach which included information technology—text messaging, emails, the use of a mobile application that participants could network, and the ‘SMART Eating’ website. In addition to technology, a ‘SMART Eating’ kit—kitchen calendar, dining tablemat, and measuring spoons—were given to participants [30]. The control group obtained brochures on nutritional education. Over six months, the study showed that both the intervention and control groups had reduced fat and salt intake levels. The net mean outcomes when comparing the intervention with the control group was −12.5 g/day for fat (*p* < 0.001) −0.51 g/day for salt (*p* < 0.001) intake [30]. The difference in both groups was their intake of fruit and vegetables; the control group, had a decrease in mean fruit and veg intake while the intervention group showed significant increases (+71.6 g/day for fruit and vegetable intake (*p* < 0.001)). 

Van Dijk et al. (2020) investigated mobile applications as a lifestyle intervention to improve healthy nutrition in women before and during early pregnancy. Risk factors in participants were measured using a Dietary Risk Score (DRS), ranging from 0 (healthy) to 9 (unhealthy) [31]. The baseline characteristics regarding nutrition in both groups showed that almost two-thirds of women reported an inadequate vegetable intake. Fruit intake was insufficient in about one-third of female participants in both groups; furthermore, approximately 10% of women reported insufficient folic acid supplementation [31]. The figures resulted in a median DRS at a baseline of 3 in both groups. Compared to participants in the control group, participants in the intervention group showed a significantly larger reduction in the DRS (β = 0.750; 95% CI 0.188–1.341), especially for vegetable intake (β = 0.550; 95% CI 0.253–0.859) (Van Dijk et al.). This study showed that there were no significant variations between groups concerning fruit consumption and folic acid supplementation. 

Han et al. (2019) reported that the difference in weight change between groups was statistically significant (mobile phone group −1.8 kg vs. control group +0.3 kg; *p* = 0.03), which was reported to be due to the mobile application allowing participants to implement behaviour change strategies effective for facilitating changes in diet and preventing relapse. Participants were able to log and track their diet, manage their daily calorie intake, and this allowed the metabolic profiles (blood pressure, glycosylated haemoglobin, total cholesterol, triglyceride, high-density lipoprotein, low-density lipoprotein, alanine aminotransferase, and visceral and subcutaneous adipose tissue; *p* < 0.05) to improve greatly [34]. Diet quality was measured on calorie consumption, expended energy and food prescription which was recommended using an embedded database based on a typical Korean diet [34]. Comparison of outcomes before and after the Health-On program showed improvements in total cholesterol, (*p*-value ≤ 0.001) triglyceride, (*p*-value ≤ 0.001), Hdl Cholesterol, (*p*-value = 0.007), LDL cholesterol, (*p*-value = 0.001), HbA1c (*p*-value ≤ 0.001) and body fat percentage, (*p*-value ≤ 0.001) [34]. Based on dietary prescription and the results such as improved markers for cholesterol and HBA1c, it is possible to say diet quality was improved by the use of mobile application [34].

Bentley et al. (2016) focused on improving the quality of the participant’s diet by using the mobile application AiperMotion 500 to inform participants of a diet suited to type 2 diabetes (a low glycaemic index diet), reducing weight loss and improving HbA1c levels [30]. The control group only received motivational email support. This study was split into three groups, group one, which received advice on diet and exercise; group two similar to group one but with the addition of the AiperMotion 500 application and group three such as group two but with the addition of email and motivational support [30]. Groups two and three could track dietary information whenever food or drink was consumed [30]. This study showed that mobile app users had a more significant reduction in HbA1c (mean range loss over study in group 2 −10.7, group 3, −5.0 compared to control group 1, +0.9) and weight (mean range loss over study in group 2 −3.3, group 3, −3.0 compared to control group 1, +0.7) compared to the control group [34]. Additionally, focus group qualitative analysis was established to obtain participant feedback on the mobile application. The feedback narratively highlighted the benefits between app use and improved diet using qualitative interviews and focus groups. 

The above findings highlight that out of the ten studies, only six included studies displayed a link between the use of mobile technology and improved diet quality. Ambrosini et al. (2018) showed no distinct differences between protein intake, fat intake, saturated fat, carbohydrate and iron density [32]. Added sugar differed between the groups in their study, but this was primarily due to misreporting [31]. In Ipjian et al. (2017), the study’s primary aim was met, and dietary sodium decreased in participants [24]. However, there were no differences between groups and diet quality scores were comparable across both the intervention and control groups throughout the study. In Wharton et al. (2014), there was a decline in HEI scores and no difference in diet quality scores between intervention and control groups [27]. Dodd et al. (2017) evaluated smartphone applications to supply lifestyle advice to pregnant women and noted that the smartphone application’s addition was not linked with any statistically notable variations in HEI scores post-intervention [28]. Of the six studies that displayed a link between diet quality and mobile applications, both Bentley et al. (2016) and Han et al. (2019) did so as an indirect consequence of their studies on mobile applications and weight loss [33,34]. Han et al. (2019) note that mobile applications allow participants to implement behaviour change strategies that facilitate changes in diet and prevent relapse [34]. The change in diet is seen in Han et al. primarily through weight loss, but additional dietary markers, such as cholesterol, show improvement [34]. In the study of Bentley et al. (2016), both weight loss and Hba1c levels improved using a dietary application supported by feedback from qualitative interviews and focus groups [33]. Van Djik et al. (2020) used dietary risk scoring to assess diet quality [31]. There was an improvement in the intervention group, which significantly reduced the DRS, especially for vegetable intake [31]. Kaur et al. (2020) showed an increase in fruit and vegetable intake using technology-enabled ‘SMART eating’ (+71.6 g/day for fruit and vegetable intake (*p* < 0.001)) [30]. Naimark et al. (2015) noted that the app users increased their diet quality score from 67 (SD 9.8) to 71 (SD 7.6; *p* < 0.001) by the end of the study [29]. Kerr et al. (2016) noted the impact of using a mobile application was a reduction in eating energy-dense, nutrient-poor foods (intervention group A, *p* < 0.001, intervention group B, *p* < 0.001), an increase in fruit servings in all participants (intervention group A, *p* = 0.03) and an increase in vegetable servings in female participants (intervention group A, *p* = 0.01) [26].

### 3.6. Secondary Findings

A secondary finding of this study was studies that were measuring mobile applications and their impact on weight loss but as an unintended consequence improved the diet quality of the participants. Wharton et al. (2014) hypothesised that smartphone app participants who record data would exhibit lower attrition rates from the study and more elevated weight loss than a control group [27]. The study showed that weight loss occurred; however, this was the case in all groups. The hypothesis was accurate regarding attrition, as app users (AP group) had 100% compliance versus the recorded dietary or pen and paper (PA) and memo group (ME). The AP group recorded significantly more whole days than the PA group (43.0 ± 2.5 and 30.7 ± 4.6 days, respectively; *p* = 0.024; effect size = 0.153) [26]. 

Han and colleagues (2019) investigated the Health-On mobile application that combines online devices (mobile app and smartwatch) and offline interventional resources, such as fitness centres and peer support. The primary aim of this research was to examine the impact of mobile technology on weight loss, which was achieved in the study [34]. This paper clearly showed that mobile applications could effectively decrease weight and other markers associated with diet and increased weight such as blood sugar and cholesterol (HbA1c (*p*-value ≤ 0.001), total cholesterol, (*p*-value ≤ 0.001)) [34].

Bentley et al. (2016) showed successful weight reduction from using the mobile application AiperMotion 500 (mean range loss over study in group 2 −3.3, group 3, −3.0 compared to control group 1, +0.7) compared to the control group [33]. The study also reported other markers associated with diet improvement and decreased weight, such as a more significant reduction in HbA1c (mean range loss over study in group 2 −10.7, group 3, −5.0 compared to control group 1, +0.9) and weight (mean range loss over study in group 2 −3.3, group 3, −3.0 compared to control group 1, +0.7) compared to the control group [33]. Both weight loss and HbA1c control were the primary outcomes of this study.

Additionally, some studies measured participants’ engagement with specific applications, and user behaviours may have meant better engagement with the mobile applications. This was carried out through participant feedback in a narrative capacity. Bentley et al. (2016) used a qualitative focus group to measure the acceptability and adherence of the mobile application. The feedback provided highlighted that three participants had issues with the applications and sought out help, while the overall feedback from the focus group reported the device was easy to use, which for some participants directly linked to adhering to a chosen diet [33]. 

While this study had no difference reported in diet quality between mobile applications and 24 h recall, Ambrosini et al. (2018) highlight the benefits of commercially developed smartphone applications which can provide valuable measures of nutrient intake and be acceptable to research participants in their study [32]. The authors also noted that the prevalence of dietary misreporting was similar in both the group using the mobile application and those using 24 h recalls [32]. Ambrosini et al. (2018) highlighted participant satisfaction with the application’s ease of use, and eighty-three percent satisfaction over using 24 h recalls in this study [32].

In the study by Dodd et al. (2017) thirty-one percent of women reported using the smartphone application during their pregnancy; while this was quite a low figure, fifty percent reported satisfaction with the usability of the application and agreed that it was likely that it aided in them to make healthier food choices [28]. The smartphone application was evaluated using a self-completed questionnaire completed by women in the intervention study who used the Lifestyle Advice plus Smartphone Application Group. 

Naimark et al. (2015) measured user behaviour regarding ease of use. Seventy-nine percent of the users noted that data entry was straightforward; ninety-three percent reported that the use of the app was straightforward [29]. According to this study, the median of the satisfaction from the app on a scale of 1–10 was 8, with an average score of 7.3 (SD 1.9) [29]. The authors highlighted that the app greatly affected the intervention group to sustain acceptable nutrition [28]. According to Naimark et al. (2015) convenience of use allowed the app users to improve their diet quality more than the control group [29].

## 4. Discussion

The purpose of this review was to systematically explore the research examining the impact of mobile technology on diet quality. The authors rigorously reviewed ten peer-reviewed articles that met the inclusion criteria to assess if using a smartphone application impacted the participants’ diet quality in the included studies. 

The primary finding of this study was that in sixty percent of studies, diet quality improved with the use of a mobile application [25,26,29,30,31]. The main change in participants’ dietary behaviour using mobile applications was increased fruit and vegetable intake [26,30,31]. The studies highlighted other benefits, such as decreased weight, improved cholesterol, improved HBA1c levels and decreased sodium intake [26,29,30,33,34]. Such improvements are typically aligned with improved health and mitigate risk factors for related dietary conditions. Results of this review indicate that mobile applications can improve the diet quality of the user. This study built on previous research which highlighted that mobile applications could be used to measure diet quality [36]. Each app gives the user the ability to catalogue a variety of nutritious foods, manage water/fluid intake, and track the recommended daily intake of vitamins and minerals [37,38,39]. Certain apps provide calorie and micronutrient content with prescribed scoring for diet quality, and apps alert the user when aims are being approached [40]. Additional features such as a barcode scanner can enhance participant experience by enabling users to quickly add products. Nudges and prompts add to user accountability and aid users in dietary modifications such as sodium reduction or calcium evaluation [40]. Adherence was also improved by using mobile applications to catalogue food which allowed users to efficiently record dietary consumption by taking pictures of food and drinks using their own device’s camera before and after eating [41]. When this study is compared to other research, it highlights that mobile applications have benefits and weaknesses.

The most notable impact on dietary outcomes this study noticed was the participants’ behaviour and how this made them interact with the mobile applications. Han et al. (2019) noted that the mobile application helped participants implement behaviour changes, allowing them to improve their diet quality. Through a focus group, Bentley et al. (2016) measured acceptability and adherence of the mobile application, which noted some both positive and negative feedback based on user experience. Naimark et al. (2015) noted that ease of use allowed the app users to improve their diet quality [29]. Behaviour and the impact on participant engagement were noted in additional research, which highlighted that a smartphone social game impacted the patient’s engagement in redirecting his/her behaviour towards better medication compliance [42]. Behaviour and the impact on participant engagement were noted in additional research, which highlighted that the use of M-health applications impacted the patient’s engagement in redirecting his/her behaviour towards better medication compliance [43]. M-health applications have the potential to treat chronic conditions by implementing behaviour changes and research has highlighted that those with a better diet quality are less likely to suffer from chronic diseases in later life [44,45].

M-health applications have the ability to offer the user support and motivation to achieve a specific goal [46]. The goal can then be numerically or visually highlighted for the user, which gives a reward for achieving a specific target [47]. When looking at the treatment of medical conditions such as type two diabetes achieving a better diet quality may address the issue, but it is often the support the individual receives which will modify their behaviour and enable a change [48]. M-health applications can provide the support and motivation an individual may need [49]. This support and motivation were studied in other M-health research and demonstrated that a reward and a goal-based system can positively affect the user [49]. Goal setting is a standard tool used in rehabilitation, in a study that introduced the concept of M-health applications to participants as a tool for rehabilitation, participants were firmly in favour of using such support tools [50,51]. Eighty-five percent of participants surveyed said they would like to sign up for the full randomised control trial, which allowed them to use an M-health application to watch videos to aid rehabilitation [50].

Similarly, M-health applications are used for dietary rehabilitation and can tackle various issues by improving the quality of the user’s diet [52]. Previous research has noted that M-health application users benefit from implementing dietary change as there is constant psychological and nutritional support with instructions to improve motivation and engagement, thereby maximising the benefits of collaborative programs [52]. This supported the secondary findings of this systematic literature review, which was the impact mobile applications have on weight loss, with three studies examining weight loss and diet with diet quality measurement resulting as an unintended consequence. In this study’s secondary findings, all three studies that examined the impact of using a mobile application to enhance weight loss noted that mobile apps promoted weight loss [27,33,34]. Additional research has highlighted that M-health acts as a successful support tool in weight management as its use allows continuity of care in out-patient settings allowing both the user and the clinician to achieve and monitor progress, respectively [52].

While community care is important, perhaps the most unique aspect of M-health is the fact that it allows the user to focus on self-monitoring of daily dietary routines. Users can digitally log their meals and calculate calories entirely by uploading food images or conducting a keyword search [53]. Studies have noted that the task of dietary logging can thus be performed as a solitary activity that requires merely the persistence and motivation of each individual [53]. Standalone M-health interventions offer an individual therapy experience due to their portability enabling self-monitoring to take place at the time of actual food and beverage consumption, allowing real-time customised feedback [54]. According to analysis, behavioural weight-loss treatment delivered via mobile technology is productive, with up to half of participants achieving clinically meaningful weight loss of 5% in standalone interventions [54]. Overall the benefits of M-health use and improving diet quality were proven to have a positive effect on health. In a 2019 study taking place over six years, higher diet quality assessed using HEI-2015 and AHEI-2010 were strongly associated with lower CVD risk [55].

Whilst the findings of this review are promising, some areas need to be analysed in greater detail. The use of mobile apps in the primary findings can improve diet quality. As highlighted by an M-health report in 2009, it is not clear if the use of mobile applications can be sustained over long periods beyond the short-term fixation with novelty [56]. A recent study highlighted an issue with the use of M-health apps and noted that almost a quarter of apps downloaded by consumers are never used a second time [57]. A significant issue faced by the long-term sustained use of M-health technology is finding significant health care partnerships to retain their users, as users do not see the value in the applications if they are not supported by health care [57]. 

There is also the issue of overdependence on M-health applications while being used for something positive such as improving diet quality which can have the caveat of being also be used to assist disordered eating patterns such as orthorexia, anorexia, or calorie restriction [58,59]. Recent research has highlighted that the use of calorie-counting and fitness-tracking technologies is worrying in relation to eating disorders [60]. According to research on disordered eating, body discontent and unease about weight and body shape tend to be high in users of pro-eating disorder websites, leading users to pay closer attention to their current weight and weight shifts and use M-health apps and web-based instruments to track their body weight and food intake [61]. Studies noted that those concerned with dieting and meal skipping are likely to have a lower diet quality rating [62]. Specific research into diet quality and eating disorders showed that diet quality scores were below expected and there were a majority of the participants had insufficient caloric intake for the HEI to be applicable [63]. Additional research is needed to make sure the value of M-health applications is not undone by misuse or M-health apps that are not supported by health care practitioners.

There are several limitations to this study. This study does not examine the facet of diet or mobile applications. Furthermore, the aim was to cover all the globally available evidence on mobile app diet quality interventions. Due to this, there was considerable heterogeneity in the results. Therefore it was impossible to say precisely how each study would have differed had they used another mobile app. The studies also used various methods to calculate diet quality, and this study does not compare the methods and how they would be transferable. The studies examined in this review do not extend past six months highlighting a need for more targeted and longer-term studies that are expressly designed to investigate the impact using mobile applications has on diet quality. To our knowledge, this is the most up-to-date systematic review evaluating the impact of mobile applications and their effects on diet quality.

## 5. Conclusions

In summary, the findings of this study have highlighted that it is key that mobile apps interact with individuals in a manner that works for them. Higher user functions have directly related to better user behaviours and higher success and improvement in diet quality. Some applications have merely provided dietary advice and improved the diet quality of participants; other applications monitor lapses and have the same effect on the user. Dietary applications that cause frustration to the user lead to complete disengagement, this can be in the form of draining the user’s battery, poor data transfer or the app failing to work as detailed [64]. For the future of dietary applications, a focus on diet quality rather than goal-setting could have more significant long-term benefits. Apps that would offer less burden on the user: taking pictures of each meal, lengthy data entry and instead allowing the user better access to food databases which introduce an array of nutrients displayed, promote better user satisfaction and engagement and overall long-term better health [64].

## Figures and Tables

**Figure 1 nutrients-14-02437-f001:**
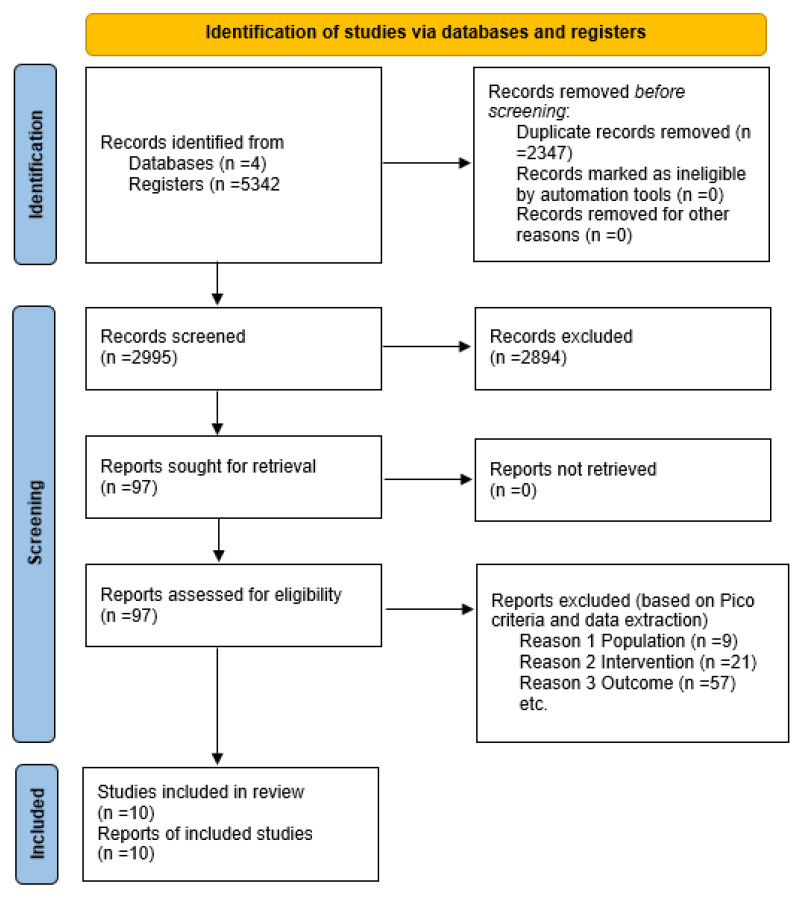
PRISMA 2020 Flow Chart of Selected Studies.

**Figure 2 nutrients-14-02437-f002:**
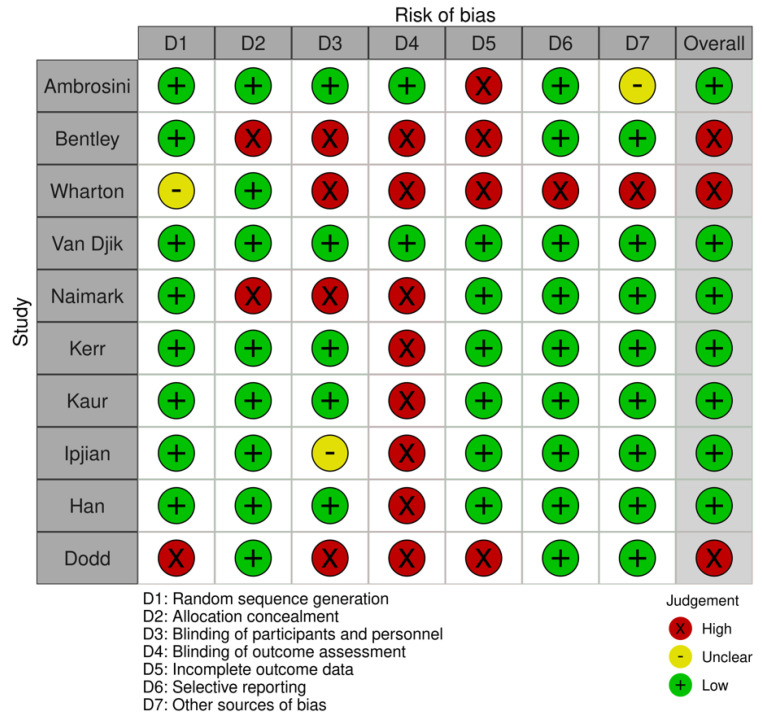
Risk of bias summary: review authors’ judgments about each risk of bias item for each included study.

**Table 1 nutrients-14-02437-t001:** Summary of study characteristics for included studies.

Author (Year)	Country	Study Design: Size, Gender and Age	Participant Characteristics	Duration	Application	Intervention	Control	Outcomes
Ambrosini et al. (2018) [32]	Australia	Short report	50 adults, 82% women mean age, 31. Majority female. Mean BMI, 22.4	4 days	Easy Diet Diary app	The Australian Calorie Counter—Easy Diet Diary smartphone app is a commercial calorie counter and food diary. Study participants completed a 4 d food diary using a modified version of the Easy Diet Diary app. The quality of diet was measured by both the intervention and control	Two 24 h recall	Average energy ratios were used to measure the quality of the participant’s diet from macronutrient fibre, iron, and calcium densities from the app and the 24 h recalls. The study using mean daily nutrient intakes from mobile applications and dietary recalls indicated no distinct differences between the app and the 24 h recalls for protein, saturated fat, carbohydrate and iron density. Added sugar intake was recorded higher in participants in the 24 h recalls.
Bentley C. L., et al. (2016) [33]	United Kingdom	Feasibility study	27 adults, Female, mean 52.9 BMI between 25 and 40 (for inclusion in the study)	39 weeks	AiperMotion 500	The intervention was a small wearable M-health device used over 12 weeks by overweight people with T2DM with the intent to lose weight and reduce their HbA1c level. This study was split into three groups, each with additional resources. Group one received advice on diet and exercise; groups two and three could track dietary information whenever food or drink was consumed.	No intervention or intervention plus weekly motivational support (group three)	This paper showed that the groups using the app benefited from weight loss and their diet management and HbA1c control. This study showed that mobile app users had a more significant reduction in HbA1c. The feedback highlighted user preference for using the mobile app to improve diet.
Dodd et al. (2017) [28]	Australia	Randomised Controlled Trial	162 Pregnant women. Approximately 43% of women were of normal BMI, 19% were overweight, and 38% were obese	26 weeks	Lifestyle Advice plus Smartphone Application	The trial evaluated the impact of a smartphone application as an adjunct to face-to-face consultations in facilitating dietary and physical activity change among pregnant women. The intervention study examined both the use of a mobile application and lifestyle advice, while the control group only received lifestyle advice.	Lifestyle Advice	No difference between quality of diet between intervention and control. All participants in the trial showed increases in milk and whole grains consumption and a decrease in sodium consumption over the trial. Although all women improved their quality of diet across pregnancy, use of the smartphone application was poor at 31%.
Han et al. (2019) [34]	Republic of Korea	Pilot Study	30 volunteers 93.3% were male and the median age was 39. Majority male. BMI 28.0 (27.2–30.3) (kg/m^2^)	12 weeks	Health-On	A weight reduction app Health-On was prescribed to the intervention group for Weight Reduction. The Health-On app has four theme pages: main, diet, physical activity, challenge and ranking. Each page allows users to see their achievements easily and maximise user convenience and app effectiveness with a simple user interface.	Did not use mobile intervention	The primary aim of this research was to examine the impact of mobile technology on weight loss, which was achieved in the study. Participants used the Health-On program to track their diet and manage their daily calorie intake; this highlighted improved diet quality when comparing outcomes before and after the Health-On program.
Ipjian et al. (2016) [25]	United States	Randomised Controlled Trial	30 adults 7 males and 23 female, mean age 34.4 majority Female. BMI 25.6 ± 4.3 kg/m^2^	4 weeks	MyFitnessPal app	Participants were instructed to reduce their sodium intake to ≤2300 mg/d by using the MyFitnessPal app to receive feedback on the sodium content of foods.	Journal tallying of foods	Participants completed a brief health history questionnaire and the Rapid Eating and Activity Assessment for Patients at the initial visit, a short, one-page, validated questionnaire to assess diet quality. At baseline, sodium intake was inversely related to diet quality. Throughout the trial, the change in diet quality scores did not differ between groups, and urinary sodium excretion decreased in the app group only compared with baseline values.
Kaur et al. (2020) [30]	India	Randomised controlled trial	732 participants 76% women, mean age 53. Majority female. Baseline 27.45 Kg/m^2^ and change −0.25 kg/m^2^	6 months	‘SMART Eating’ intervention	The intervention included information technology SMS, email, social networking app and ‘SMART Eating’ website, interpersonal communication, and distribution of a ‘SMART Eating’ kit—kitchen calendar, dining table mat, and measuring spoons. The intervention was executed at the household level over six months.	Pictorial pamphlet on the dietary recommendations of National Institute of Nutrition, India, with information written in Hindi language	Primary outcomes were changes in mean dietary intakes of fat, sugar, salt, and fruit and vegetables, there was a secondary improvement of changes in BMI, blood pressure, haemoglobin, FPG, and serum lipids. This study used M-health and showed improvement in diet quality concerning their intake in fruit and vegetables.
Kerr et al. (2016) [26]	Australia	Randomised Controlled Trial	247 participants 162 women and 85 men. Mean age (years) 24.2 ± 3.2, 23.7 ± 3.4, 25.0 ± 3.5 in groups a, b and c respectively. Majority female. Dietary feedback only group showed the weight change from baseline = −1.75 kg and BMI change, BMI (*p* = 0.01)	6 months	Mobile food record App (mFR)	(A). Dietary feedback and weekly text messages, (B) dietary feedback. Dietary intake was assessed using a mobile food record App (mFR) where participants captured images of foods and beverages consumed over 4-days at baseline and post-intervention.	Control did not receive any dietary feedback or text messages.	This study showed improvement in diet quality related to the use of mFR application. This included uptake of fruit and veg and a decrease in EDNP foods in men and SSB in women and a reduction in body weight
Naimark et al. (2015) [29]	Israel	Randomised Controlled Trial	85 participants 64% women and 36% men. The mean age was 47.9 (SD 12.3) years. BMI was 26.2 (SD 3.9)	14 weeks	Web-based app	Access to the app without any face-to-face support.	The control subjects continued their standard lifestyle	The study noted that the app users increased their diet quality score by the end of the study. Based on guidelines from a 16-item questionnaire based on Parmenter’s general nutrition knowledge questionnaire for adults, diet quality was measured using an online self-reported questionnaire. The improvement in diet quality scores was no difference between light and heavy users of the mobile application.
van Dijk et al. (2020) [31]	Netherlands	Randomised Controlled Trial	218 participants Women between aged 18 and 45, median age 30.6 (5.3) 30.7 (5.7) years. BMI Not measured	24 weeks	The Smarter Pregnancy program.	Intervention group received personal online coaching based on identified inadequate intakes of vegetables, fruits, and folic acid supplement	No coaching or application	Dietary risk score (DRS), improved in the women using the mobile application, this was due to larger intake of fruit and vegetables
Wharton et al. (2014) [27]	United States	Randomised Controlled Trial	57 participants Age years 43.7 ± 3.5, 41.5 ± 4.0, 40.8 ± 3.8. (Group a, b and c respectively). Gender, 12 male, 35 female.	8 week	Lose it	The intervention group used the mobile app Lose It (group 1)	Groups 2 and 3 used the memo feature on a smartphone, or a traditional paper-and-pencil method, respectively.	Weight loss was the measurement of this study, however, it was noted the participants using the app had an increase in the consumption of fruit and veg and the research suggests that mobile applications improve diet quality.

Abbreviations: BMI: Body mass index, T2DM: type two diabetes mellitus, HbA1c: Glycated haemoglobin, app: application/mobile application, mg/d: milligrams per day, kg/m^2^: Kilogram-Meter Squared, SMS: short message service, FPG: fasting plasma glucose, mFR: mobile food record, EDNP: energy-dense nutrient-rich, SSB: sugar-sweetened beverages, SD: standard deviation, DRS: Dietary risk score.

## Data Availability

Not applicable.

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
