# Peer review of "Usage of Mobile Applications or Mobile Health Technology to Improve Diet Quality in Adults"

_nutrients, 2022, doi:10.3390/nu14122437_

Round 1

Reviewer 1 Report

This is a well-written review about the impact of mobile applications on diet quality.

I doubt whether the study of Ambrosine et al (ref 32) should be included in this review.  As far as I know this study is a validity study and not an intervention. According to me improvement of dietary quality was not the subject of this study.

Two minor points in Table 1: Participants characteristics

Ipjian et al: BMI 25.6 4.3 (what is the 4,.3? Is this the SD?)

and Kaur et al: BMI -0,25 kg/m2. Is this the BMI change

Reviewer 2 Report

Systematic Review

On the use of mobile applications or mobile health technology to improve diet quality in adults

-          In the current study, only 10 studies were used which were suitable for evaluation of the impact of using mobile applications or mobile health technology to improve diet quality in adults. 6 studies evaluated the using of mobile applications or mHealth interventions to highlight dietary health changes, while 4 studies used to reduce weight or blood sugar levels

-          The study concludes that mobile applications may be an effective way to improve diet quality in adults; however, there is a need for more targeted and longer-term studies that are expressly designed to investigate the impact using mobile applications has on diet quality.

-          This research is a novel study, it is an attempt to evaluate the effect of using mobile technology to improve diet quality in adults. But it needs to collect more study to evaluate the effect of using mobile technology to improve diet quality in adults, because these 10 studies cannot make decision and cannot generalized their findings.

Important notes:

Title:

I am suggesting changing of the title to be usage of mobile applications or mobile health technology to improve diet quality in adults

In the abstract:

Some typing errors should be corrected such as (4studies)

3.5. Study outcomes

Line 261 and 262;

Ambrosini et al. (2018) tested the feasibility of using smartphone technology for dietary assessment over using 24hr recalls, testing mean (SD) daily intakes of total energy.

-conclusion needs to be written proficiently according to the findings of the study.

- References also are not well written according to the journal instructions.
